# Enhancing Abstractiveness of Summarization Models through Calibrated Distillation

**Hwanjun Song, Igor Shalyminov, Hang Su, Siffi Singh, Kaisheng Yao, Saab Mansour**
AWS AI Labs
{hwanjuns, shalymin, shawnsu, siffis, kaishey, saabm}@amazon.com

## Abstract

Sequence-level knowledge distillation reduces the size of Seq2Seq models for more efficient abstractive summarization. However, it often leads to a loss of abstractiveness in summarization. In this paper, we propose a novel approach named DisCal to enhance the level of abstractiveness (measured by $n$-gram overlap) without sacrificing the informativeness (measured by ROUGE) of generated summaries. DisCal exposes diverse pseudo summaries with two supervision to the student model. Firstly, the best pseudo summary is identified in terms of abstractiveness and informativeness and used for sequence-level distillation. Secondly, their ranks are used to ensure the student model to assign higher prediction scores to summaries with higher ranks. Our experiments show that DisCal outperforms prior methods in abstractive summarization distillation, producing highly abstractive and informative summaries. Code is publicly available at https://c1kj.short.gy/discal.

## 1 Introduction

Text summarization is the task of generating a concise and condensed summary of a source document while preserving its most important information (Gupta and Gupta, 2019). Unlike extractive summarization, which involves selecting and concatenating sentences from the original document (Nallapati et al., 2017), *abstractive* summarization is a sequence-to-sequence (Seq2Seq) problem that can generate novel phrases and sentences that were not present in the original document (Nallapati et al., 2016; Paulus et al.; Fan et al., 2018; Gupta and Gupta, 2019). Recent advances in large pre-trained language models have greatly accelerated summarization modeling progress (Lewis et al., 2020; Zhong et al., 2022), but there are still significant concerns for the large language model's real use cases due to the slow inference speed under a production-level environment.

Figure 1: Summaries generated from the models without using knowledge distillation, "w.o. Distil"; using sequence-level distillation, "Seq-level Distil" (Zhang et al., 2022a); and using Calibrated Distillation ("DisCal", ours) on CNNDM data. Fragments from the input are color-coded to indicate overlap: green, yellow, and red for over three, five, and ten tokens, respectively.

*Knowledge distillation* is a widely used technique for compressing a large model into a smaller one for faster inference with minimal performance loss (Ba and Caruana, 2014; Hinton et al., 2015; Chen et al., 2020). A prominent direction for abstractive summarization is known as *sequence-level* distillation (Kim and Rush, 2016; Zhang et al., 2022a). This method involves generating a pseudo summary for each training document using a teacher model, and training a student model on pairs of training documents and their corresponding pseudo summaries. Compared to methods that rely on word-level information, such as minimizing the cross-entropy loss between teacher and student prediction distributions (Gou et al., 2021), this approach enables the student to better mimic the teacher model's generation at the sequence level.

Despite the high ROUGE (Lin, 2004) score achieved through sequence-level distillation, we argue that the pseudo summary generated by the teacher model exacerbates the student model's ten-

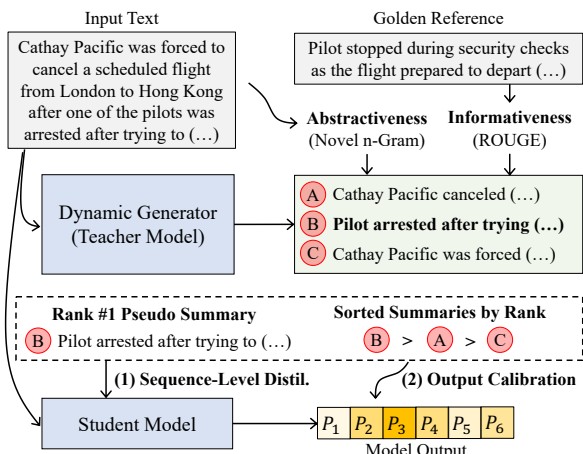

Figure 2: Overview of DisCal: The teacher efficiently transfers its knowledge through two approaches: firstly, employing *sequence-level distillation*, utilizing the best pseudo summary in terms of abstractiveness and informativeness, and secondly, applying *output calibration*, making higher-ranked summaries receive correspondingly higher predicted scores.

dency to copy continuous text segments from the source documents, thus intensifying the problem of *copy bias* during summary generation. As seen in Figure 1, relying solely on the teacher's pseudo summaries for distilling knowledge, without utilizing gold summaries, compels the student model to generate *extractive-like* summaries due to the inherent copy bias (see the Seq-level Distil). Thus, the level of abstractiveness remains limited, hindering the student model's capacity to produce truly informative and coherent abstractive summaries.

This trade-off between informativeness and abstractiveness is a significant challenge in abstractive summarization (Zhang et al., 2018; Lin and Ng, 2019), yet it has not been addressed within the context of knowledge distillation.

In this paper, we present the notion of *calibrated distillation*, which entails distilling knowledge by precisely calibrating the pseudo summaries provided by the teacher. Our proposed method, referred to as **DisCal**, as illustrated in Figure 2, leverages the teacher model as a *dynamic summary generator* that generates diverse pseudo summaries for each input text. To enhance the diversity, we dynamically manipulate the attention temperature of the teacher model throughout the distillation process, mitigating copy-bias by exposing numerous summaries to the student model.

To evaluate the quality of the pseudo summaries, we employ a ranking system based on two factors: *informativeness*, which is assessed using the ROUGE score, and *abstractiveness*, which is mea-

sured by the ratio of novel $n$-grams in a summary that are not present in the input text. For knowledge distillation, we select the best pseudo summary in terms of the two factors to supervise the student model through sequence-level distillation. Additionally, the ranking of the summaries is used to calibrate the student model, ensuring that it assigns higher prediction scores to summaries with higher ranks. By doing these, DisCal enhances the level of abstractiveness and improves the ROUGE score, showing promising potential even when the gold summaries in training data are less abstractive.

Our contributions are threefold: (1) we unveil the issue of reduced abstractiveness in current sequence-level distillation, (2) we introduce a novel approach called calibrated distillation to achieve better informativeness-abstractiveness trade-off, and (3) the proposed DisCal method surpasses existing state-of-the-art approaches in abstractive summarization distillation on three news and dialogue summarization datasets, namely CNNDM (Hermann et al., 2015), XSUM (Narayan et al., 2018), and SAMSum (Gliwa et al., 2019).

## 2   Related Work

Large pre-trained Seq2Seq models have emerged as the de facto standard for abstractive summarization due to their exceptional performance and versatility. They excel in capturing the salient information from documents through various techniques. For instance, T5 (Raffel et al., 2020) predicts corrupted text spans, BART (Lewis et al., 2020) employs denoising auto-encoding, PEGASUS (Zhang et al., 2020) identifies the most summary-worthy sentences, and DialogLED (Zhong et al., 2022) employs window-based denoising.

Due to the high computational cost associated with large models, there has been a surge of research focused on compressing these models (Gou et al., 2021; Frantar et al., 2023). One prominent approach in this field is known as knowledge distillation, which involves training a smaller student model to mimic the predictions of a larger teacher model by minimizing the difference between the teacher and student predictions (Ba and Caruana, 2014; Hinton et al., 2015; Chen et al., 2020). Particularly, in the context of abstractive summarization distillation with Seq2Seq models, Kim and Rush (2016) proposed the sequence-level knowledge distillation approach which involves training a student model using pseudo summaries generated by the teacher model using beam search decoding. On

the other hand, Shleifer and Rush (2020) proposed the shrink and fine-tune (SFT) framework. This approach involves removing certain layers from the teacher model to create a smaller student model, which is then fine-tuned using gold summaries. In a recent study, Zhang et al. (2022a) introduced the method PLATE, which aims to smooth attention distributions of teacher models during pseudo summary generation and then fine-tune the shrunken student model with them.

In addition, there is an interesting line of work called model calibration (Liu and Liu, 2021; Zhang et al., 2022b; Liu et al., 2022), where they leverage different candidate summaries to calibrate the model's predictions to overcome the problem of exposure bias (Bengio et al., 2015). In contrast to prior research, our work focuses on the previously overlooked problem of decreased abstractiveness when distilling summarization models. We propose a solution called Calibrated Distillation, which achieves a high level of informativeness and abstractiveness using a smaller model.

## 3 Preliminary

### 3.1 Seq2Seq Abstractive Summarization

Abstractive summarization aims at generating a *concise* summary of a given document or text using *novel* phrases and sentences. The objective of abstractive summarization is to learn a neural Transformer model $\Theta$[1] that receives a source document $X = \{x_1, x_2, \ldots, x_{|X|}\}$ and generates its appropriate summary $Y = \{y_1, y_2, \ldots, y_{|Y|}\}$, where $x_t$ and $y_t$ are the word token in the document and its summary at time $t$, respectively.

For this objective, the Seq2Seq Transformer can be trained to maximize the conditional probability:

$$p(Y|X;\Theta) = \prod_{t=1}^{|Y|} p(y_t|Y_{<t}, X; \Theta), \quad (1)$$

where the notation $Y_{<t}$ represents all word tokens preceding the position $t$. Consequently, the model is updated to minimize the negative log-likelihood loss (NLL) for each pair of input document $X$ and its gold summary $Y^*$ in the training data:

$$\ell_{\mathrm{NLL}}(X, Y^*) = -\frac{1}{|Y^*|} \sum_{t=1}^{|Y^*|} \log p(y_t^*|X, Y_{<t}^*; \Theta). \quad (2)$$

[1] We mainly focus on Seq2Seq Transformer models (Vaswani et al., 2017) for abstractive summarization.

### 3.2 Sequence-level Knowledge Distillation

Let $\Theta_t$ and $\Theta_s$ be the teacher and student models, where the student must be smaller in size compared to the teacher. Given the teacher model $\Theta_t^*$ trained by Eq. (2), the teacher's output distribution for the document $D$ is approximated by the *pseudo* summary $\tilde{Y} = \{\tilde{y}_1, \tilde{y}_2, \ldots, \tilde{y}_{|\tilde{Y}|}\}$, which is the output from running beam search with the teacher model (Kim and Rush, 2016), where the summary with the highest beam score is selected. Therefore, the student model is trained to mimic the teacher's summary generation process by minimizing the NLL loss on the teacher-generated summary $\tilde{Y}$, *i.e.*, $\ell_{\mathrm{NLL}}(X, \tilde{Y})$. The gold summary $Y^*$ in training data is no longer used in sequence-level knowledge distillation.

## 4 Methodology

We introduce a new distillation method Dis-Cal (Abstractive Summarization **Dis**tillation with **Cal**ibration) in this section. Briefly speaking, the dynamic summary generator (in Section 4.1) provides a list of feasible pseudo summaries for each document and the calibrated distillation (in Section 4.2) tunes the student model to output the summary with high informativeness and abstractiveness.

### 4.1 Dynamic Summary Generator

In sequence-level knowledge distillation, using a *single* deterministic pseudo-summary generated by the trained teacher model is sub-optimal. This approach limits the exposure of the model to diverse valid summaries for a given input document (Liu et al., 2021a), leading to reduced abstractivenss. Additionally, it easily propagates incorrect predictions from the teacher model to the student model due to overly confident predictions (Guo et al., 2020; Liang et al., 2022). Consequently, this can lead to poor performance in generating accurate abstractive summaries.

To address the issues, we utilize the teacher model as a *dynamic summary generator* (in Figure 2), enabling it to generate diverse pseudo summaries in real-time during the distillation process. This is achieved by employing the diverse beam search technique (Vijayakumar et al., 2018) and randomly re-scaling its attention temperature within a predefined range. Manipulating attention weights in all attention modules is recognized for its effectiveness in mitigating the copy bias in the generated summary (Zhang et al., 2022a). There-

fore, we randomly re-scale the attention temperature of the Transformer model,

$$\text{Attention}(Q, K, V) = \text{softmax}(\frac{QK^\top}{k\sqrt{d}})V, \quad (3)$$

where $Q, K, V$ are linear projections of hidden states of each Transformer layer; and $k$ is a randomly drawn re-scaling factor from the uniform distribution $\text{U}(1, \gamma)$; and $\gamma$ is the maximum value for re-scaling. Thus, the teacher model generates a list of $n$ pseudo summaries via diverse beam search, $\tilde{\mathcal{Y}} = \{\tilde{Y}_1, \tilde{Y}_2, \ldots, \tilde{Y}_n\}$, and these summaries differ even for the same document according to which re-scaling factor is chosen. Examples of pseudo summaries can be found in Appendix A.

### 4.2 Calibrated Distillation

We introduce a new concept of distillation named *calibrated distillation*, which is built on the standard sequence-level distillation pipeline but differs in terms of considering two more aspects: (1) we utilize the gold summary to identify the most reliable pseudo summary from the summary list $\tilde{\mathcal{Y}}$ and (2) we calibrate the student model's output such that it can generate the summary with high informativeness and abstractiveness.

Specifically, given the list of pseudo summaries $\tilde{\mathcal{Y}}$, DisCal evaluates and ranks the $n$ summaries in the list in terms of informativeness and abstractiveness. We define the calibration score to evaluate the ranks by employing ROUGE and novel n-gram scores, which are respectively for informativeness and abstractiveness[2], as in Definition 4.1.

**Definition 4.1.** Let $\tilde{\mathcal{Y}} = \{\tilde{Y}_1, \tilde{Y}_2, \ldots, \tilde{Y}_n\}$ be the list of $n$ pseudo summaries for an input document. Then, the informativeness score $s_{\text{info}}(\tilde{Y}_i)$ for the $i$-th pseudo summary is the average of ROUGE-1, ROUGE-2, and ROUGE-L F1 scores on the gold summary $Y^*$, and the abstractiveness score $s_{\text{abs}}(\tilde{Y}_i)$ for the $i$-th pseudo summary is the average of novel 1-gram, 3-gram, and 5-gram scores with respect to the input document $X$. Hence, the *calibration score* is formulated by the weighted sum of the two scores normalized over $n$ pseudo summaries in the list $\tilde{\mathcal{Y}}$ as:

$$s_{\text{calib}}(\tilde{Y}_i) = (1 - \lambda)\bar{s}_{\text{info}}(\tilde{Y}_i) + \lambda\bar{s}_{\text{abs}}(\tilde{Y}_i),$$

$$\text{s.t. } \bar{s}_{\text{info}}(\tilde{Y}_i) = s_{\text{info}}(\tilde{Y}_i)/\sum_{j=1}^{n} s_{\text{info}}(\tilde{Y}_j)$$

$$\text{and } \bar{s}_{\text{abs}}(\tilde{Y}_i) = s_{\text{abs}}(\tilde{Y}_i)/\sum_{j=1}^{n} s_{\text{abs}}(\tilde{Y}_j), \quad (4)$$

where $\lambda$ is the balancing term of $s_{\text{info}}$ and $s_{\text{abs}}$, adjusting the importance of the two factors. □

Accordingly, we now obtain a list of *ranked pseudo summaries* $\tilde{\mathcal{Y}}' = \{Y_1', Y_2', \ldots, Y_n'\}$ such that $\forall_{i<j} s_{\text{calib}}(Y_i') < s_{\text{calib}}(Y_j')$. We use this updated list for calibrated knowledge distillation.

Firstly, the summary $Y_n'$ is selected as the best summary among all pseudo summaries in $\tilde{\mathcal{Y}}'$ since it exhibits the highest calibration score $s_{\text{calib}}$. Hence, we employ $Y_n'$ as the *target* summary for guiding the student model through sequence-level knowledge distillation. The student model learns from the teacher's knowledge by minimizing a modified NLL loss. In this case, the loss equation remains identical to Eq. (2), but the target is substituted with the rank 1 pseudo summary, denoted as $\ell_{\text{NLL}}(X, Y_n')$. By incorporating the ROUGE score into the assessment process, we ensure that the selected summary has a high level of informativeness.

Secondly, motivated by the work that leverages the order of candidate summaries (Zhang et al., 2022b; Liu et al., 2022), we encourage the student model's prediction output such that it assigns higher estimated probabilities to high ranked summaries. For a given pseudo summary $\tilde{Y}$, the length-normalized estimated log-probability (Liu et al., 2022) by the student model is formulated as:

$$f(\tilde{Y}) = \frac{1}{|\tilde{Y}|^\alpha} \sum_{t=1}^{|\tilde{Y}|} \log p(\tilde{y}_t | X, \tilde{Y}_{<t}; \Theta_s), \quad (5)$$

where $\tilde{Y} = \{\tilde{y}_1, \tilde{y}_2, \ldots, \tilde{y}_{|\tilde{Y}|}\}$ and $\alpha$ is a length penalty hyperparameter similarly used for beam search. Then, our calibration loss is formulated by using the margin based pairwise ranking loss (Hopkins and May, 2011) as:

$$\ell_{\text{Calib}}(X, \tilde{\mathcal{Y}}') = \sum_{i<j} \max(0, f(\tilde{Y}_j) - f(\tilde{Y}_i) + m_{ij}), \quad (6)$$

where $m_{ij} = (j - i) * m$ represents the margin multiplied by the difference in rank between two pseudo summaries. Intuitively, this encourages the student model's log-probability $f(\tilde{Y}_j)$ to be greater than $f(\tilde{Y}_i)$ since $s_{\text{calib}}(\tilde{Y}_j) > s_{\text{calib}}(\tilde{Y}_i)$, thereby generating summaries with high levels of informativeness and abstractiveness.

---

[2]Novel $n$-gram score is the ratio of $n$-grams in the summary that do not appear in the input document, which is widely used to measure the abstractiveness in the literature (Liu and Lapata, 2019; Zhang et al., 2022a; Dreyer et al., 2023)

| Dataset | # Training | # Validation | # Testing | novel 1-gram | novel 3-gram | novel 5-gram | Task |
|---------|-----------|--------------|-----------|--------------|--------------|--------------|------|
| CNNDM | 287,113 | 13,368 | 11,490 | 18.05 | 76.05 | 88.87 | News Summarization |
| XSUM | 204,045 | 11,332 | 11,334 | 85.40 | 99.78 | 99.98 | News Summarization |
| SAMSum | 14,732 | 818 | 819 | 59.07 | 93.93 | 98.84 | Dialogue Summarization |

Table 1: Summary of datasets. Novel $n$-gram scores are computed on pairs of input documents and their gold summaries. A higher $n$-gram score indicates that the gold summaries in the test set are more abstractive.

As a result, the student model is trained by combining the two loss objectives for sequence-level knowledge distillation and model output calibration, respectively, as:

$$\ell_{\text{DisCal}} = \eta * \underbrace{\ell_{\text{NLL}}(X, Y'_n)}_{\text{Seq-level KD}} + \underbrace{\ell_{\text{calib}}(X, \tilde{\mathcal{Y}})}_{\text{Output Calib.}}, \quad (7)$$

where $\eta$ is the weight for the NLL loss.

## 5 Evaluation

**Datasets.** We evaluate DisCal on three widely-used abstractive summarization datasets: two news summarization datasets of CNN/DailyMail (Hermann et al., 2015) and XSUM (Narayan et al., 2018); and a dialogue summarization dataset of SAMSum (Gliwa et al., 2019).

- The CNNDM dataset comprises online news articles sourced from the CNN and DailyMail websites, each accompanied by corresponding highlight summaries for reference.

- The XSUM dataset contains online articles from BBC News with single sentence summaries, which are more abstractive than those in CNNDM (Dreyer et al., 2023).

- The SAMSum dataset contains messenger-like conversations with summaries written by linguists. Unlike CNNDM and XSUM, SAMSum involves dialogue data that includes more than two participants.

The detailed statistics of the datasets can be found in Table 1. It is important to note that each dataset exhibits varying levels of abstractiveness in its gold summaries. XSUM and SAMSum exhibit a very high level of abstractiveness compared to CNNDM, probably because their summary length is very short; mostly a single sentence.

**Teacher and Student Models.** Following the literature (Zhang et al., 2022a), we consider BART Large (Lewis et al., 2020), one of the widely used Seq2Seq Transformer architectures for abstractive summarization. The BART Large model is trained on the entire dataset with gold summaries as a *teacher* model. Then, we configure two *student* models with identical Transformer encoder layers to the teacher, but they differ in the number of decoder layers: BART 12-6 and BART 12-3, with six and three decoding layers, respectively. Referring to the SFT pipeline (Shleifer and Rush, 2020), the student models are initialized from the 12-encoder-layer/12-decoder-layer teacher. The two student models copy the full encoder from the teacher model. But, the decoder of BART 12-6 is copied from the $\{0, 2, 4, 6, 8, 10\}$ decoder layers of the teacher, while the decoder of BART 12-3 from the $\{0, 5, 11\}$ decoder layers. This initialization is simple but effective since it eliminates the need for separately pre-training the two student models.

In Appendix D, we validate the generalizability of DisCal on a different state-of-the-art Seq2Seq model, DialogLED (Zhong et al., 2022).

**Algorithms.** We compare DisCal with the three prior knowledge distillation approaches, namely *shrink and then fine-tune (SFT)* (Shleifer and Rush, 2020) and two sequence-level knowledge distillation methods, *Seq-Distil* (Kim and Rush, 2016) and PLATE (Zhang et al., 2022a). The SFT method trains the student model on pairs of documents and their corresponding gold summaries without using pseudo summaries. On the other hand, the other two methods only rely on the pseudo summary generated by the teacher model using beam search decoding; PLATE is different from Seq-Distil in terms of scaling up the teacher's attention temperature in pseudo summary generation. We re-implement all compared methods and train them in the same environment using eight NVIDIA V100 GPUs and Pytorch 1.13.1 (Paszke et al., 2019).

**Implementation Details.** Similar to recent studies (Rohde et al., 2021; Zhang et al., 2022a), we train BART Large (teacher model) using the Adam optimizer (Kingma and Ba, 2014) with an initial learning rate of $5e$-5 and a label smoothing of $0.1$. The teacher model is trained for $20,000$ steps on CNNDM and XSUM with a weight decay of $0.001$ and a batch size of $64$, while $5,000$ steps on SAMSum with a weight decay of $0.1$ and a batch size of $16$. We use the same training configuration on the two student models for Seq-distil and PLATE.

| Method | Informativeness | | | Abstractiveness | | |
| | ROUGE-1 | ROUGE-2 | ROUGE-L | Novel 1-gram | Novel 3-gram | Novel 5-gram |
|---|---|---|---|---|---|---|
| Teacher Model: BART Large with 406M parameters | | | | | | |
| BART Large (Lewis et al., 2020) | 44.80 | 21.47 | 41.80 | 7.46 | 36.74 | 52.27 |
| Student Model: BART 12-6 with 306M parameters | | | | | | |
| SFT (Shleifer and Rush, 2020) | 44.73 | 21.37 | 41.76 | 6.86 | 35.65 | 51.78 |
| Seq-Distil (Kim and Rush, 2016) | 44.14 | 21.33 | 41.16 | 5.52 | 28.26 | 42.24 |
| PLATE (Zhang et al., 2022a) | 45.33 | 22.13 | 42.52 | 6.87 | 35.26 | 50.92 |
| DisCal (ours) | 46.76 | 22.58 | 44.07 | 10.77 | 56.76 | 76.62 |
| Student Model: BART 12-3 with 255M parameters | | | | | | |
| SFT (Shleifer and Rush, 2020) | 44.47 | 21.31 | 41.71 | 7.33 | 37.39 | 55.14 |
| Seq-Distil (Kim and Rush, 2016) | 44.23 | 21.22 | 41.71 | 4.69 | 24.98 | 38.40 |
| PLATE (Zhang et al., 2022a) | 44.78 | 21.65 | 42.03 | 6.54 | 32.72 | 48.45 |
| DisCal (ours) | 46.16 | 21.92 | 43.62 | 10.91 | 56.99 | 76.88 |

Table 2: Comparison on CNNDM data for news summarization. We reproduced all the methods. The reproduced BART Large shows a better ROUGE-1 score than the original implementation performance of 44.16.

| Method | ROUGE-1 | ROUGE-2 | ROUGE-L | Novel 1-gram | Novel 3-gram | Novel 5-gram |
|---|---|---|---|---|---|---|
| Teacher Model: BART Large with 406M parameters | | | | | | |
| BART-Large (Lewis et al., 2020) | 45.35 | 22.50 | 37.50 | 37.73 | 93.08 | 98.34 |
| Student Model: BART 12-6 with 306M parameters | | | | | | |
| SFT (Shleifer and Rush, 2020) | 44.84 | 21.42 | 36.35 | 36.76 | 92.94 | 98.49 |
| Seq-Distil (Kim and Rush, 2016) | 44.20 | 20.86 | 35.67 | 35.44 | 90.64 | 97.11 |
| PLATE (Zhang et al., 2022a) | 44.71 | 21.43 | 36.53 | 35.42 | 91.32 | 97.67 |
| DisCal (ours) | 45.24 | 21.91 | 37.25 | 36.88 | 92.68 | 98.30 |
| Student Model: BART 12-3 with 255M parameters | | | | | | |
| SFT (Shleifer and Rush, 2020) | 43.68 | 20.91 | 36.26 | 37.45 | 93.72 | 98.92 |
| Seq-Distil (Kim and Rush, 2016) | 43.48 | 20.36 | 35.43 | 35.56 | 90.56 | 97.18 |
| PLATE (Zhang et al., 2022a) | 43.78 | 20.86 | 36.11 | 35.61 | 91.41 | 97.81 |
| DisCal (ours) | 44.30 | 21.14 | 36.73 | 37.35 | 92.98 | 98.43 |

Table 3: Comparison on XSUM data for news summarization. We reproduced all the methods. The reproduced BART Large shows a better ROUGE-1 score than the original implementation performance of 45.14.

As for our hyperparameters, we tune them on validation sets. The maximum value $\gamma$ for re-scaling in Eq. (3), the balancing term $\lambda$ for the calibration score in Eq. (4), and the weight for NNL loss in Eq. (7) are respectively set at 2.0, 0.2, and 0.01 on CNNDM; 1.5, 0.2, and 1.0 on XSUM; and 1.5, 0.2, and 0.1 on SAMSum. The number of pseudo summaries $n$ per document is set at 6. The detailed implementation including hyperparameter settings for all methods are provided in Appendix B.1.

Regarding inference, we apply beam search following the convention (Lewis et al., 2020; Zhang et al., 2022a). We set the beam size, length penalty, minimum length, and maximum length to 4, 2.0, 55, and 142 on CNNDM; 6, 2.0, 10, and 62 on XSUM; 6, 2.0, 11, and 62 on SAMSum. For evaluation, we use ROUGE F1 and novel $n$-gram scores as the informativeness and abstractiveness metrics. Refer to Appendix B.2 for details.

## 5.1 Results on News Summarization

Tables 2 and 3 summarize the results obtained from two news summarization datasets. The first block shows the performance of the teacher model (BART Large), while the second and third blocks include the results achieved by the two student models (BART 12-6 and BART 12-3) trained using four different knowledge distillation methods.

In general, DisCal exhibits the best performance in terms of informativeness and abstractiveness in both datasets. Particularly, DisCal shows significant performance improvements on the CNNDM dataset. This dataset, as indicated in Table 1, exhibits a low level of abstractiveness in its gold summaries, leaving ample room for improvement. The two students models with DisCal even surpass the performance of their teacher model with large margin. On the other hand, the two existing sequence-level distillation methods, Seq-Distil and PLATE sacrifice the level of abstractiveness compared to the teacher model and SFT. For XSUM, we observe the similar trend of exhibiting the highest ROUGE while maintaining better novel $n$-gram scores than the two sequence-level distillation methods. A less significant improvement to abstractiveness comes from short length summaries of XSUM.

| Method | Informativeness | | | Abstractiveness | | |
|---|---|---|---|---|---|---|
| | ROUGE-1 | ROUGE-2 | ROUGE-L | Novel 1-gram | Novel 3-gram | Novel 5-gram |
| Teacher Model: BART Large with 406M parameters | | | | | | |
| BART Large (Lewis et al., 2020) | 53.24 | 28.65 | 49.23 | 46.60 | 83.52 | 94.03 |
| Student Model: BART 12-6 with 306M parameters | | | | | | |
| SFT (Shleifer and Rush, 2020) | 52.52 | 27.49 | 48.21 | 47.88 | 85.12 | 95.12 |
| Seq-Distil (Kim and Rush, 2016) | 52.16 | 27.22 | 47.94 | 44.40 | 80.31 | 92.22 |
| PLATE (Zhang et al., 2022a) | 53.11 | 28.34 | 49.03 | 45.25 | 81.19 | 92.62 |
| DisCal (ours) | 53.66 | 28.96 | 49.97 | 47.85 | 84.72 | 94.65 |
| Student Model: BART 12-3 with 255M parameters | | | | | | |
| SFT (Shleifer and Rush, 2020) | 47.38 | 23.63 | 43.72 | 52.74 | 89.01 | 97.44 |
| Seq-Distil (Kim and Rush, 2016) | 50.68 | 26.08 | 46.84 | 46.28 | 81.53 | 93.10 |
| PLATE (Zhang et al., 2022a) | 50.73 | 26.34 | 46.90 | 47.96 | 82.98 | 93.67 |
| DisCal (ours) | 51.65 | 26.72 | 48.08 | 48.74 | 84.79 | 94.92 |

Table 4: Comparison on SAMSum data for dialogue summarization. We reproduced all the methods.

| Model | # Param | Inference Latency | | |
|---|---|---|---|---|
| | | CNNDM | XSUM | SAMSum |
| BART Large | 406M | 700ms | 304ms | 385ms |
| BART 12-6 | 306M | 377ms | 212ms | 190ms |
| BART 12-3 | 255M | 275ms | 129ms | 125ms |

Table 5: Number of parameters and latency (milliseconds per document) on V100 GPU with batch size 1.

| Component | Informativeness | | Abstractiveness |
|---|---|---|---|
| | ROUGE-1 | ROUGE-2 | Novel 5-gram |
| SFT (Default) | 44.47 | 21.31 | 55.14 |
| $\ell_{\text{NLL}}(\lambda = 0.0)$ | 45.32 | 21.99 | 52.83 |
| $\ell_{\text{NLL}}(\lambda = 0.2)$ | 45.31 | 21.95 | 55.24 |
| $\ell_{\text{Calib}}(\lambda = 0.2)$ | 2.52 | 0.00 | 0.0 |
| $\ell_{\text{NLL}} + \ell_{\text{Calib}}$ | 46.16 | 21.92 | 76.88 |

Table 6: Ablation study by adding each loss component.

| Coefficient | Informativeness | | Abstractiveness |
|---|---|---|---|
| | ROUGE-1 | ROUGE-2 | Novel 5-gram |
| $\lambda = 0.2$ | 46.16 | 21.92 | 76.88 |
| $\lambda = 0.4$ | 44.82 | 19.96 | 89.56 |
| $\lambda = 0.6$ | 43.75 | 18.71 | 92.78 |

Table 7: Varying the terms $\lambda$ which balances between abstractiveness and informativeness according to Eq. (4).

| Number | Informativeness | | Abstractiveness |
|---|---|---|---|
| | ROUGE-1 | ROUGE-2 | Novel 5-gram |
| $n = 3$ | 45.49 | 21.60 | 72.66 |
| $n = 6$ | 46.16 | 21.92 | 76.88 |
| $n = 9$ | 46.29 | 21.75 | 83.56 |
| $n = 12$ | 46.42 | 21.30 | 86.67 |

Table 8: Varying the number of pseudo summaries.

## 5.2 Results on Dialogue Summarization

Table 4 shows the results on the dialogue SAMSum dataset. DisCal maintains its performance dominance compared to the other distillation approaches. Similar to the CNNDM dataset, BART 12-6 with DisCal exhibits ROUGE and novel $n$-gram scores higher than its teacher model.

## 5.3 Inference Latency

Table 5 summarizes the number of trainable parameters and inference latency of BART models we used. By reducing the number of decoder layers, the parameter size decreases from 406M to 255M. Notably, as the decoder is the most computationally intensive component during inference due to auto-regressive decoding, the student models demonstrate a significantly lower inference latency compared to the teacher model. Specifically, BART 12-6 and BART 12-3 achieve inference speed improvements of $1.43 – 2.03$ and $2.36 – 3.08$ times faster than BART Large, respectively. Despite faster inference speed, the student models enhanced with DisCal exhibit comparable or superior perfor-

mance to BART Large in generating informative and highly abstractive summaries, as demonstrated from the results presented in Tables 2, 3, and 4.

## 5.4 Detailed Analysis on Main Component

We perform a detailed analysis of DisCal on CNNDM data using BART 12-3.

### 5.4.1 Loss Component Ablation Study

We perform an ablation study on DisCal by gradually adding each loss component on top of the SFT method. The results are shown in Table 6. Firstly, we use the NLL loss in Eq. (2) with $\lambda = 0.0$, where we consider only the informativeness score $s_{\text{info}}$ for selecting the best summary without utilizing the calibration loss from Eq. (6). In this setting, although the ROUGE score exhibits a considerable improvement, the novel 5-gram score drops. Secondly, when increasing the $\lambda$ value to 0.2, where the best pseudo summary is selected by considering both informativeness and abstractiveness, the ROUGE and novel 5-gram scores are both improved. Next, by utilizing both the NLL loss and the calibration loss, we observe further enhancements in the ROUGE and novel 5-gram scores due to their synergistic

| Method | Summary | Document |
|---|---|---|
| Gold | The wildfire started in Miami-Dade County on Sunday. By Monday night, it had grown to nearly 2,000 acres . The fire was 50% contained, officials said . | Parts of Miami-Dade County's skyline was hidden from view Monday as smoke from a growing 1,850-acre wildfire loomed over portions of the Florida county. What started as a nonthreatening and seemingly shrinking grass fire on Sunday, consuming fewer than 100 acres according to Miami-Dade Fire Rescue Battalion Chief Al Cruz, grew to be more than 10 times that within the next 24 hours. By Monday night, the fire had burned nearly 2,000 acres and was 50% contained, the fire department said. High temperatures and gusty winds helped the fire spread, State Forester Jim Karels said. Several fire units and a helicopter with the capacity to drop 400 gallons of water at a time were battling the blaze, Cruz said. "The Florida Forest Service and Miami-Dade Fire Rescue have worked around the clock to protect Southwest Miami-Dade County," Florida Agriculture Commissioner Adam H. Putnam said in a statement. Early Monday night, officials were considering road closures, and one school, Lincoln Marti, was evacuated as a precaution, according to the Fire Department. |
| STF (21.1 / 63.6) | The fire has burned nearly 2,000 acres and is 50% contained. High temperatures and gusty winds helped the fire spread, state forester says. One school is evacuated as a precaution. The fire is 50 percent contained. Florida agriculture commissioner: The fire is around the clock | |
| Seq-Distil (33.6 / 45.9) | A grass fire in Miami-Dade County has burned nearly 2,000 acres and is 50% contained. High temperatures and gusty winds helped the fire spread, State Forester Jim Karels says. Several fire units and a helicopter are battling the blaze. | |
| PLATE (22.3 / 36.1) | The fire has burned nearly 2,000 acres and is 50% contained. High temperatures and gusty winds helped the fire spread, State Forester Jim Karels says. Several fire crews and a helicopter with the capacity to drop 400 gallons of water at a time are battling the blaze. | |
| DisCal (42.4 / 71.4) | The fire has burned nearly 2,000 acres and is 50% contained, fire officials say. The fire started as a nonthreatening grass fire on Sunday. A school is evacuated as a precaution. Firefighters are battling the blaze in Miami-Dade County. | |

Table 9: Example of summaries generated from four different methods including gold summary on CNNDM using BART 12-3. Fragments that overlap the input document by five or more words are marked in red. The values in parenthesis under the method name are the informativeness and abstractiveness scores, ($s_{\text{info}}$ / $s_{\text{abs}}$).

| Method | Informativeness | | Abstractiveness |
|---|---|---|---|
| | ROUGE-1 | ROUGE-2 | Novel 5-gram |
| DisCal | 46.16 | 21.92 | 76.88 |
| + Self Distil | 47.50 | 23.58 | 67.57 |

Table 10: Improvement with self-calibrated distillation.

| Method | Con | Coh | Rel | Flu |
|---|---|---|---|---|
| BART Large | 4.91 | 4.67 | 4.06 | 2.99 |
| STF | 4.90 | 4.57 | 3.88 | 2.97 |
| Seq-Distil | 4.79 | 4.55 | 3.99 | 2.97 |
| PLATE | 4.82 | 4.55 | 3.92 | 2.98 |
| DisCal (ours) | 4.82 | 4.53 | 4.04 | 2.96 |

Table 11: Human-like evaluation using G-EVAL on consistency (Con), coherence (Coh), relevance (Rel), and fluency (Flu), where BART Large is the teacher model of the four distilled models.

impact. However, using only the calibration loss does not work as there is no supervision from the NLL loss for summary generation.

### 5.4.2 Balancing Term for Calibration Score

The hyperparameter $\lambda$ in Eq. (4) balances the importance between informativeness and abstractiveness in evaluating pseudo summaries. The higher the $\lambda$ value, the greater the importance of abstractiveness in the score. Table 7 demonstrates how the ROUGE and novel 5-gram scores are affected by adjusting the $\lambda$ value. With increasing $\lambda$ values, we observe a trade-off between the levels of informativeness and abstractiveness in the summaries. Placing excessive weight on abstractiveness compromises the level of informativeness in the summary; the abtractiveness improves while the informativeness drops considerably.

### 5.4.3 Number of Pseudo Summaries

Intuitively, increasing the number $n$ of pseudo summaries from the dynamic teacher model provide more performance gain with DisCal. Table 8 shows how increasing $n$ affects the performance of DisCal. As the number increases, DisCal generates better summaries in terms of the ROUGE-1 and novel 5-gram scores, while the ROUGE-2 score begins to drop slightly when $n$ is greater than 6. Therefore, in general, having more pseudo summaries helps generate highly abstract summaries without sacrificing much informativeness.

### 5.4.4 Qualitative Analysis

Table 9 presents an example of generated summaries on the test data from CNNDM. The two approaches, Seq-Distil and PLATE, generate more informative summaries compared to SFT, as they exhibit high informativeness score $s_{\text{info}}$, However, they sacrifice the abstractiveness score $s_{\text{abs}}$, which is lower than that of SFT. This indicates that they copy a large portion of the summaries from the input document (see the red fragments shown in table 9). In contrast, DisCal is not only robust against copy bias but also achieves a very high informativeness score compared to other methods. We provide additional examples in Appendix C.

### 5.4.5 Self-Calibrated Distillation

Our method has potential in leveraging enhanced student model as self-teacher for subsequent training. Here, we set $\lambda = 0.0$ in this bootstrapping experiment as the student model has already been trained with DisCal ($\lambda = 0.2$). Table 10 presents the results before and after self-calibration using BART 12-3 trained with DisCal on CNNDM. While ROUGE scores got improved, novel 5-gram scores decrease. Thus, a $\lambda$ value greater than 0.0 is still necessary to maintain high abstractiveness.

| Method | Informativeness | | | Abstractiveness | | |
|---|---|---|---|---|---|---|
| | ROUGE-1 | ROUGE-2 | ROUGE-L | Novel 1-gram | Novel 3-gram | Novel 5-gram |
| SFT w. Back-translation | 41.82 | 17.29 | 38.56 | 17.65 | 67.27 | 84.44 |
| Seq-Distil w. Back-translation | 41.16 | 17.16 | 37.93 | 17.12 | 64.89 | 82.29 |
| PLATE w. Back-translation | 42.36 | 17.90 | 39.27 | 17.53 | 66.83 | 84.22 |
| DisCal wo. Back-translation | 46.76 | 22.58 | 44.07 | 10.77 | 56.76 | 76.62 |

Table 12: Impact of using back-translation to existing distillation methods using BART 12-6 on CNNDM.

## 5.5 Human-like Evaluation using GPT-4

We conduct human-like evaluation using G-EVAL (Liu et al., 2023). This is a novel LLM-based evaluation approach employing GPT-4, outperforming all prior automated methods and also displaying a substantial Spearman correlation of 0.513 with human scores in summarization tasks. We use exactly the same prompt suggested from the authors, employing a scale of 1 (worst) to 5 (best) for consistency, coherence, and relevance, and 1 (worst) to 3 (best) for fluency. Table 11 shows the results of four distillation models using BART 12-6, including their teacher BART Large, on CNNDM.

Our analysis yields the two insights. Firstly, all distillation methods have slight impact on consistency, coherence, relevance, and fluency; up to 0.18 difference compared to the teacher. This likely stems from the use of teacher-generated pseudo summaries, which effectively prevents performance divergence in student models. Secondly, DisCal enhances abstractiveness while maintaining high consistency. This is achieved through the integration of ROUGE (between pseudo and gold summary) in summary selection and output calibration, ensuring the student model to retain crucial contents from the gold summary during training.

## 5.6 Comparison with Paraphrasing

Paraphrasing is a simple method for enhancing the abstractiveness of provided summaries (Li et al., 2022; Zhou and Bhat, 2021). Hence, we evaluate one of the paraphrasing techniques known as back-translation. We utilize Amazon Translate [2], a fluent and accurate machine translation service and explore a form of back translation: English → German → English. Table 12 summarizes the impact of back-translation on CNNDM when it is applied to the model output generated by SFT, Seq-Distil, and PLATE using BART 12-6.

The results demonstrate that the back-translation effectively enhances abstractiveness of existing methods, yet it noticeably reduces informativeness

(i.e., ROUGE) compared to not using it. In contrast, our approach, Discal, strikes a more favorable balance between informativeness and abstractiveness by the proposed calibrated distillation, resulting in improvements in both aspects.

## 6 Conclusion

We propose DisCal, an improved knowledge distillation method that leverages diverse candidate summaries generated by teacher model. By evaluating and ranking pseudo summaries during distillation, DisCal chooses the best summaries in terms of informativeness and abstractiveness, and this enhances model predictions based on output calibration. Experiments on three summarization datasets demonstrate that DisCal produces summaries with a higher level of abstractiveness as well as informativeness.

## Limitations

DisCal introduces additional training overhead. Generating pseudo summaries from the teacher model involves beam search decoding, which is computationally intensive compared to simple teacher forcing. However, this computational overhead in training phase does not affect inference in testing, i.e. DisCal does not require any changes in inference.

Regarding the training overhead, some recent studies show that beam decoding can be expedited using techniques such as early exiting (Liu et al., 2021b; Schuster et al., 2022) and parallel decoding (Santilli et al., 2023). These research show great potential on alleviating the burden associated with beam decoding during training.

## Ethics Statement

This paper focuses on general abstractive summarization and knowledge distillation, introducing a novel calibrated distillation method that produces summaries with high levels of abstractiveness and informativeness. To evaluate our method, we use public benchmark datasets, i.e. CNNDM, XSUM, and SAMSum. Therefore, we do not anticipate any negative ethical and social impact.

[2]https://aws.amazon.com/translate/

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
