# OpenReview forum: "Enhancing Abstractiveness of Summarization Models through Calibrated Distillation"
_EMNLP/2023/Conference — EMNLP 2023 Findings_

### Official Review · Reviewer_U7yc · 2023-07-23

**Soundness:** 3

**Excitement:**

3: Ambivalent: It has merits (e.g., it reports state-of-the-art results, the idea is nice), but there are key weaknesses (e.g., it describes incremental work), and it can significantly benefit from another round of revision. However, I won't object to accepting it if my co-reviewers champion it.

**Missing References:**

Paraphrasing tools/methods are not covered.

**Paper Topic And Main Contributions:**

The paper proposes a new teacher-student method for abstractive summarization that improves abstractiveness of summaries while keeping the quality of the original model (BART in this case).

**Questions For The Authors:**

l. 010 There is a typo in this line

l.395-396 This is confusing, is the BART model trained from scratch here? It is unlikely. Probably, the model is tuned - in this case, what is the base model?

Table 2: There is very significant improvement in abstractiveness, but it is not clear whether or not Rouge scores are higher in a significant way. What were the results of significance test?

l.445-450 This is a description of results, but what is the reason for them? Why does abstrativeness improve better for 1-grams than for 3- and 5-grams? It looks like simple word replacement from these results on XSUM. What is the difference from CNN/DM? Why does it work better there?

**Reasons To Accept:**

The results do show improve in abstractiveness for 1-grams. Summary quality does not suffer.

**Reasons To Reject:**

This method is tested on BART underlying model, but will it work for other models?
Also, in most of the cases, the improvement in abstractiveness happens for 1-grams only (which means word replacement), and not for larger n-grams. Will the ordinary word replacement methods work better? Also, would it not be more productive to use a summarization => paraphrasing pipeline that will give an excellent abstractiveness?

**Reproducibility:**

4: Could mostly reproduce the results, but there may be some variation because of sample variance or minor variations in their interpretation of the protocol or method.

**Reviewer Confidence:**

4: Quite sure. I tried to check the important points carefully. It's unlikely, though conceivable, that I missed something that should affect my ratings.

---

> ### Author Rebuttal · Authors · 2023-08-28
>
> We sincerely appreciate the reviewer's constructive comments and feedback on our manuscript. Throughout this rebuttal, we have strived to address your concerns and sincerely hope that our response meets your expectations.
>
> ---
>
> > **Clarification on Main Contribution**
>
> The goal of this work is to achieve high quality abstractive summaries using smaller models for faster inference. Prior distillation methods compromise abtractiveness for increased informativeness due to the inherent trade-off. In contrast, the paraphrasing method sacrifices informativeness for improved abstractiveness (refer to the response addressing W4). Our contribution entails not only substantial improvement on informativeness, but also an increase in the level of abstractiveness.
>
> ---
>
> > **W1.** *This method is tested on BART underlying model, but will it work for other models?*
>
> Thanks for pointing out this important issue. Due to the lack of space, we presented the results with an alternative model in Table 13 of Appendix D, as mentioned in Lines 374-376. Specifically, we evaluated our method  on the CNNDM dataset using DialogLED-Base [a, b], which is one of the state-of-the-art summarization models.
>
> |                                                    | ROUGE-1 | ROUGE-2 | ROUGE-L | Novel 1-gram | Novel 3-gram | Novel 5-gram |
> | -------------------------------------------------- | ------- | ------- | ------- | ------------ | ------------ | ------------ |
> | Teacher Model: DialogLED Base with 162M parameters |
> | DialogLED                                          | 43.13   | 20.18   | 40.26   | 4.99         | 24.97        | 38.55        |
> | Student Model: DialogLED 6-3 with 133M parameters  |
> |  SFT                                                | 42.83   | 19.90   | 40.14   | 5.83         | 30.00        | 45.87        |
> |  Seq-Distil                                         | 42.60   | 19.70   | 39.64   | 3.81         | 18.34        | 29.51        |
> |  Plate                                            | 41.69   | 19.27   | 39.21   | 7.36         | 33.39        | 49.52        |
> |  DisCal (ours)                                      | 44.56   | 20.86   | 42.19   | 10.24        | 54.42        | 75.15        |
>
> The results demonstrate that, similar to the results with the BART model, the improvement by DisCal on CNNDM is consistently high; it improves the Rouge score significantly while enhancing the level of abstractiveness in summaries.
>
> [a] DialogLM: Pre-trained Model for Long Dialogue Understanding and Summarization, AAAI 2022
>
> [b] Huggingface checkpoint for fine-tuning: https://huggingface.co/MingZhong/DialogLED-base-16384
>
> ---
>
> > **W2.** *In most of the cases, the improvement in abstractiveness happens for 1-grams only and not for larger n-grams. What is the difference from CNN/DM? Why does it work better there?*
>
> The variance in abstractiveness improvement comes from the intrinsic differences among summarization datasets, as demonstrated in Table 1. CNNDM data features multi-sentence summaries with relatively lower abstractiveness, whereas XSUM and SAMSum encompass short single-sentence summaries with higher abstractiveness.
>
> In XSUM and SAMSum, existing methods already attain high novel 3- and 5-gram scores (80.31 to 98.92 out of 100). Here, a novel 5-gram score of 98.92 indicates that 98.92% of 5-grams are not present in the input document. Thus, there is limited room for improvement compared to novel 1-grams.
>
> In contrast, CNNDM presents ample room for improvement due to less abstractive gold summaries in the training data. As a result, our method substantially enhances even novel 3- and 5-grams.
>
> ---
>
> > **W3**. *The smaller improvement of 3- and 5-grams on XSUM looks like simple word replacement.*
>
> Although a marginal difference exists in novel 3- and 5-gram, this is not a simple word replacement scenario. To illustrate this point, we provide an example of summaries generated for the same document in the XSUM dataset.
>
> This is the summary from an existing method, ${\rm P_{LATE}}$:
> ```
> The US Navy has imposed a curfew on its base in Okinawa after a US Navy officer was arrested on suspicion of drink-driving.
> ```
>
> This is the summary from our method, DisCal:
> ```
> The US Navy has banned its personnel from drinking after a female sailor was arrested on suspicion of drink-driving on the Japanese island of Okinawa.
> ```
>
> The two summaries are clearly more than word replacements. However, they exhibit similar novel 3- and 5-gram scores due to their use of distinct n-grams from the input article, as follows:
>
> |              | ROUGE-1 | ROUGE-2 | ROUGE-L | Novel 1-gram | Novel 3-gram | Novel 5-gram |
> | ------------ | ------- | ------- | ------- | ------------ | ------------ | ------------ |
> | Plate        | 60.37   | 31.37   | 52.83   | 17.39        | 85.71        | 94.73        |
> | DisCal (our) | 65.45   | 45.28   | 65.45   | 20.00        | 86.95        | 95.23        |
>
> Therefore, our method creates a summary with a considerable distinct structure, striving for improved Rouge scores but also improved novel n-gram scores.
>
> ---
>
> > **W4.** *Paraphrasing tools/methods are not covered. Will paraphrasing pipeline give an excellent abstractiveness?*
>
> Thank you for bringing up an important topic for discussion. According to our literature survey [c, d], there have been several ways of doing paraphrasing, namely back-translation, rule-based method, and thesauruses.
>
> As per your suggestion, we tested one of the paraphrasing technique known as back-translation, which can readily enhance abstractiveness of summaries. We utilized Amazon Translate [e], a fluent and accurate machine translation service, and explored a form of back translation: English→German→English for paraphrasing.
>
> * CNNDM using BART 12-6
>
> |                       | ROUGE-1 | ROUGE-2 | ROUGE-L | Novel 1-gram | Novel 3-gram | Novel 5-gram |
> | --------------------- | ------- | ------- | ------- | ------------ | ------------ | ------------ |
> | SFT                  | 44.73   | 21.37   | 41.76   | 6.86         | 35.65        | 51.78        |
> | \+ back-translatation | 41.82   | 17.29   | 38.56   | 17.65        | 67.27        | 84.44        |
> | Seq-Distil            | 44.14   | 21.33   | 41.16   | 5.52         | 28.26        | 42.24        |
> | \+ back-translatation | 41.16   | 17.16   | 37.93   | 17.12        | 64.89        | 82.29        |
> | Plate                | 45.33   | 22.13   | 42.52   | 6.87         | 35.26        | 50.92        |
> | \+ back-translatation | 42.36   | 17.90   | 39.27   | 17.53        | 66.83        | 84.22        |
> | DisCal (ours)         | 46.76   | 22.58   | 44.07   | 10.77        | 56.76        | 76.62        |
>
> * XSUM using BART 12-6
>
> |                       | ROUGE-1 | ROUGE-2 | ROUGE-L | Novel 1-gram | Novel 3-gram | Novel 5-gram |
> | --------------------- | ------- | ------- | ------- | ------------ | ------------ | ------------ |
> | SFT                   | 44.84   | 21.42   | 36.35   | 36.76        | 92.94        | 98.49        |
> | \+ back-translatation | 41.63   | 17.88   | 33.56   | 39.72        | 94.97        | 99.25        |
> | Seq-Distil            | 44.20   | 20.86   | 35.67   | 35.44        | 90.64        | 97.11        |
> | \+ back-translatation | 41.07   | 17.47   | 32.98   | 38.97        | 93.89        | 98.81        |
> | Plate                 | 44.71   | 21.43   | 36.53   | 35.42        | 91.32        | 97.67        |
> | \+ back-translatation | 41.04   | 17.67   | 33.29   | 38.83        | 94.16        | 98.96        |
> | DisCal (ours)         | 45.24   | 21.91   | 37.25   | 36.88        | 92.68        | 98.30        |
>
> The results demonstrate that the back-translation effectively enhances abstractiveness, yet it noticeably reduces informativeness (i.e., Rouge).  In contrast, our approach, Discal, strikes a more favorable balance between informativeness and abstractiveness by the proposed calibrated distillation, resulting in improvements in both aspects. We will add the literature survey on paraphrasing and these experimental results in the final version.
>
> [c] Data augmentation approaches in natural language processing: A survey, AI Open 2022
>
> [d] Paraphrasing generation: a survey of the state of the art, EMNLP 2021
>
> [e] https://aws.amazon.com/translate/
>
> ---
>
> > **Q1**. *Is the BART model trained from scratch here? What is the base model of BART?*
>
> We apologize for any confusion caused. The BART models were fine-tuned from the Hugging Face checkpoint for BART Large (https://huggingface.co/facebook/bart-large). We will include this detail in the paper.
>
> ---
>
> > **Q2.** *There is very significant improvement in abstractiveness in Table 2 (CNNDM results). But it is not clear whether or not Rouge scores are higher in a significant way. What were the results of significance test?*
>
> In Table 2, our improved Rouge scores surpass existing distillation methods significantly. Specifically, a recent study, ${\rm P_{LATE}}$ [f], noted that enhancements of  0.84-1.01 in Rouge-1/-2/-L on CNNDM are significant (versus to the Shrink-and-Finetune using BART 12-6). In the same setup, we not only achieved much higher Rouge gain of 1.21-2.31, but also successfully improved abstractiveness by resolving their trade-off.
>
> Moreover, we provide the standard errors for Table 2 to assess the statistical significance of Rouge improvements by our method. We conducted the experiments three times using different seeds, and the standard error for each method was: 0.030 — 0.055 for STF, 0.031 — 0.055 for Seq-Distil, 0.027 — 0.035 for Plate, and 0.027 — 0.053 for Discal (ours). Therefore, the improvement in Rouge of Table 2 is statistically significant, given the much smaller standard error.
>
> [f] Attention Temperature Matters in Abstractive Summarization Distillation, ACL 2022
>
> ---
>
> > **Other Update**
>
> As per request from other reviewers, we conducted additional evaluation on other metrics (e.g., consistency, relevance, fluency). DisCal’s impact on these metrics appears modest despite the generation of more abstractive summaries. Please refer to our responses addressing W1, W1-1, and W1-2 from Reviewer XooF, if you are interested.

---

### Official Review · Reviewer_XooF · 2023-08-03

**Soundness:** 4

**Excitement:**

3: Ambivalent: It has merits (e.g., it reports state-of-the-art results, the idea is nice), but there are key weaknesses (e.g., it describes incremental work), and it can significantly benefit from another round of revision. However, I won't object to accepting it if my co-reviewers champion it.

**Paper Topic And Main Contributions:**

This paper proposes a novel sequence-level knowledge distillation method for abstractive summarization.
By generating diverse summaries using a teacher model and ranking them regarding informativeness (ROUGE) and abstractiveness (novel n-gram), the proposed method improves the level of abstractiveness, as well as the informativeness of generated summaries.
Experimental results demonstrated that their method outperforms other distillation methods, and the student model achieves higher informativeness and abstractiveness than the teacher model.

**Questions For The Authors:**

It is somewhat surprising that the ROUGE scores of student models outperform those of teacher models.
This result indicates that training with pseudo summaries achieves higher ROUGE scores than training with gold summaries, though the pseudo summaries are selected using ROUGE scores against the gold summaries.
What is the rationale for this result?

**Reasons To Accept:**

The paper is well-written and clearly states the merit of this approach (improvement of informativeness and abstractiveness) by providing sufficient empirical evidence.
The proposed method is simple and reasonable, and sufficient experiments are conducted with appropriate settings (e.g., the ablation study, fair experimental settings, and hyperparameter sensitivity analysis).

**Reasons To Reject:**

Conventional summarization studies evaluate the output summary using several criteria other than informativeness and abstractiveness. Evaluation from other criteria (e.g., factuality, coherency, fluency) should also be conducted in this study.
Specifically, evaluating factuality is important for this study, as teacher models sometimes generate hallucinations, and higher novel n-gram often leads to more hallucinated contents. Selected higher ROUGE & novel n-gram pseudo summaries may contain lower factual contents, and the trained student model may generate hallucinated contents. The factuality of the generated summaries should be compared with that of the student model trained with gold summaries.

**Reproducibility:**

4: Could mostly reproduce the results, but there may be some variation because of sample variance or minor variations in their interpretation of the protocol or method.

**Reviewer Confidence:**

4: Quite sure. I tried to check the important points carefully. It's unlikely, though conceivable, that I missed something that should affect my ratings.

---

> ### Author Rebuttal · Authors · 2023-08-28
>
> We sincerely appreciate the reviewers' constructive comments and feedback on our manuscript. The weaknesses of our paper are discussed as follows.
>
> ---
>
> > **W1.** *Evaluation from other criteria (e.g., factuality, coherency, fluency) should also be conducted in this study.*
>
> Thank you for bringing up an important issue. We completely agree with your point because the improvement of abstractiveness may affect the other metrics you mentioned.
>
> In response, we conducted human-like evaluation using G-EVAL [a] on CNNDM. This is a novel LLM-based evaluation approach employing gpt-4, outperforming all prior automated methods, displaying a substantial Spearman correlation of 0.514 with human scores in summarization tasks. The [same prompts](https://github.com/nlpyang/geval/tree/main/prompts/summeval) were used, employing a scale of 1 (worst) to 5 (best) for consistency, coherence, and relevance, and 1 (worst) to 3 (best) for fluency.
>
> | Metric                                   | Consistency | Coherence | Relevance | Fluency |
> | ---------------------------------------- | ----------- | --------- | --------- | ------- |
> | Teacher: BART Large with 406M parameters |
> | BART Large                               | 4.91        | 4.67      | 4.06      | 2.99    |
> | Student: BART 12-6 with 306M parameters  |
> | SFT                                      | 4.90        | 4.57      | 3.88      | 2.97    |
> | Seq-Distil                               | 4.79        | 4.55      | 3.99      | 2.97    |
> | Plate                                    | 4.82        | 4.55      | 3.92      | 2.98    |
> | DisCal (ours)                            | 4.82        | 4.53      | 4.04      | 2.96    |
>
> Generally, all distillation methods have slight impact on consistency, coherence, relevance, and fluency (up to 0.18 difference compared to the teacher). This likely stems from the use of teacher-generated pseudo summaries, which effectively prevents performance divergence in student models.
>
> [a] G-Eval: NLG Evaluation using GPT-4 with Better Human Alignment, arXiv 2023
>
> ---
>
> > **W1-1.** *Evaluating factuality is important for this study, as teacher models sometimes generate hallucinations, and higher novel n-gram often leads to more hallucinated contents. Selected higher Rouge & novel n-gram pseudo summaries may contain lower factual contents, and the trained student may generate hallucinated contents.*
>
> Based on the above results, DisCal’s impact on generating hallucinated content appears modest, as evidenced by the close consistency score between the teacher (BART Large trained on gold summaries) and DisCal.
>
> This is achieved through the integration of Rouge (between pseudo and gold summary) in summary selection and output calibration, ensuring the student model to retain crucial contents from the gold summary during training. Note that in Eq. (4), a much higher weight is assigned to the informativeness score than the abstractiveness score with the lambda=0.2 (we used in experiments). This prevents the student from choosing abstractive summaries lacking gold summary content, thus mitigating potential factual inaccuracies.
>
> ---
>
> > **W1-2.** *The factuality of the generated summaries should be compared with that of the student model trained with gold summaries.*
>
> The SFT method is a student trained on gold summaries using the Shrink-and-Finetuning framework. As shown in the table, they have comparable factual consistency. However, our approach excels in abstractiveness (novel n-grams) and informativeness (Rouge) compared to the SFT method (refer to Table 2).
>
> In addition, we conducted pairwise comparison with ChatGPT [b] using the following template:
>
> ```
> Given a new article, which summary is factually better abstractive summary? Answer "Summary 0" or "Summary 1". You do not need to explain the reason.
>
> Article: {article}
> Summary 0: {summary 0}
> Summary 1: {summary 1}
> ```
>
> We got a win count of 102 and 98 for the STF method and our method, measured on SummEval data (100 examples sampled from CNNDM) twice by changing the order of summaries for the fair comparison. This indicates that training with pseudo summaries having higher Rouge and novel n-gram scores yields factual consistency comparable to training with gold summaries.
>
> [b] Human-like Summarization Evaluation with ChatGPT, arXiv 2023
>
> ---
>
> >**Q1.**  *It is somewhat surprising that the Rouge scores of student models outperform those of teacher models. This result indicates that training with pseudo summaries achieves higher Rouge scores than training with gold summaries, though the pseudo summaries are selected using Rouge scores against the gold summaries. What is the rationale for this result?*
>
> We incorporated two design principles aimed at enhancing the Rouge, which are not considered in teacher's training.
>
> * **Exposure to Diverse Valid Summaries:**  In training data, the gold summary represents only one potential summary among many. Consequently, the “training with gold summary pipeline” restricts the model’s exposure to a variety of valid summaries. In our approach, the pseudo summary’s structure and word usage differ even for the same document, as we dynamically adjust the teacher model’s attention temperature. Here, the Rouge score aids in identifying the most valid pseudo summary from N candidates, retaining crucial contents in the gold summary.
>
> * **Use of rank information:** we utilize rank information derived from multiple pseudo summaries, different from the conventional supervised loss associated with a single gold summary. Utilizing the calibration score in Definition 4.1, we assess the ranks of N summaries, and instruct the student model to prioritize the summary with a higher rank over the one with a lower rank.
>
> These design principles prove to be more advantageous for longer summaries, where more diverse summaries can be generated from the teacher. This explains why our method benefits CNNDM more than XSUM, which features short single-sentence summaries. We will clarify this rational in the final version.

---

### Official Review · Reviewer_1fso · 2023-08-10

**Soundness:** 4

**Excitement:**

3: Ambivalent: It has merits (e.g., it reports state-of-the-art results, the idea is nice), but there are key weaknesses (e.g., it describes incremental work), and it can significantly benefit from another round of revision. However, I won't object to accepting it if my co-reviewers champion it.

**Paper Topic And Main Contributions:**

This work is on enhancing knowledge distillation approaches for abstractive summarization models. It is an important research direction aiming to obtain high quality abstractive summarization models, while having small models with fast inference-time. In particular, this work tackles the tradeoff between informativeness and abstractivness in distilled models. Current knowledge distilled summarization models loose abstractive capabilities.

The authors propose a new method "Abstractive Summarization Distillation with Calibration" and show superior performance over their teacher models and other distillation models in terms of informativeness and abstractivness (as measured by ngram overlaps).

Their approach builds on the Shrink-and-finetune framework, but improves it by introducing 1) a teacher model as dynamic summary generator and 2) callibrated distillation.

**Questions For The Authors:**

The summarization datasets differ in their abstractiveness. Did you try modifying the balancing hyperparameter lambda depending on the dataset? It seems 0.2 is good for CNN/DM as shown in 5.4.1, but probably for XSUM the value is different?

**Reasons To Accept:**

- the proposed approach outperforms other distilled models and even the teacher model on many datasets achieving high abstractiveness while maintaining high informativeness.
- the experiments and analyses are extensive. Models are evaluated on typical datasets. Further, all components of their approach (calibration and distillation using dynamic summary generation) are analyzed in ablation studies. Different hyperparameters are evaluated and the results are analyzed qualitatively.
- the paper is well structured and clearly written, making it easy to follow

**Reasons To Reject:**

- the advances over the existing approaches are minimal on some evaluation metrics.
- Ignore: This is now fixed in the rebuttal (informativeness and abstractiveness are the only evaluation metrics. Other metrics like fluency, coherence, faithfulness are ignored.)

**Reproducibility:**

4: Could mostly reproduce the results, but there may be some variation because of sample variance or minor variations in their interpretation of the protocol or method.

**Reviewer Confidence:**

4: Quite sure. I tried to check the important points carefully. It's unlikely, though conceivable, that I missed something that should affect my ratings.

**Typos Grammar Style And Presentation Improvements:**

I would suggest highlighting (bold) the highest scores in the evaluation tables.

---

> ### Author Rebuttal · Authors · 2023-08-28
>
> We sincerely appreciate the reviewers' constructive comments and feedback on our manuscript. The weaknesses of our paper are discussed as follows.
>
> ---
> > **W1.** *The advances over the existing approaches are minimal on some evaluation metrics.*
>
> We interpret the minimal improvement as novel n-gram performance on XSUM. This is attributed to highly abstractive gold summaries (as shown in Table 1). In this case, balancing increased abstractiveness without sacrificing Rouge scores is more challenging. Although the novel n-gram improves less significantly, the improvement in Rouge surpasses existing methods. Specifically, a recent method, ${\rm P_{LATE}}$ [a], noted that enhancements of 0.25-0.39 in Rouge-1/2/L on XSUM is significant (versus to the Shrink-and-Finetune using BART 12-6). In the same setup, we not only achieved higher Rouge gain of 0.40-0.90, but also successfully improved abstractiveness by resolving their trade-off.
>
> [a] Attention Temperature Matters in Abstractive Summarization Distillation, ACL 2022
>
> ---
>
> > **W2**. *Informativeness and abstractiveness are the only evaluation metrics. Other metrics like fluency, coherence, faithfulness are ignored.*
>
> Thank you for bringing up an important issue. We completely agree with your point because the improvement of abstractiveness may affect the other metrics you mentioned.
>
> In response, we conducted human-like evaluation using G-EVAL [b]. This is a novel LLM-based evaluation approach employing gpt-4, outperforming all prior automated methods, displaying a substantial Spearman correlation of 0.514 with human scores in summarization tasks. The [same prompts](https://github.com/nlpyang/geval/tree/main/prompts/summeval) were used, employing a scale of 1 (worst) to 5 (best) for consistency, coherence, and relevance, and 1 (worst) to 3 (best) for fluency.
>
> | Metric                                   | Consistency | Coherence | Relevance | Fluency |
> | ---------------------------------------- | ----------- | --------- | --------- | ------- |
> | Teacher: BART Large with 406M parameters |
> | BART Large                               | 4.91        | 4.67      | 4.06      | 2.99    |
> | Student: BART 12-6 with 306M parameters  |
> | SFT                                      | 4.90        | 4.57      | 3.88      | 2.97    |
> | Seq-Distil                               | 4.79        | 4.55      | 3.99      | 2.97    |
> | Plate                                    | 4.82        | 4.55      | 3.92      | 2.98    |
> | DisCal (ours)                            | 4.82        | 4.53      | 4.04      | 2.96    |
>
> Our analysis yields the following insights:
>
> * All distillation methods have slight impact on consistency, coherence, relevance, and fluency (up to 0.18 difference compared to the teacher). This likely stems from the use of teacher-generated pseudo summaries, which effectively prevents performance divergence in student models.
> * DisCal (our method) enhances abstractiveness while maintaining high consistency. This is achieved through the integration of Rouge (between pseudo and gold summary) in summary selection and output calibration, ensuring the student model to retain crucial contents from the gold summary during training.
>
> We will add these results and discussions in the final version.
>
> [b] G-Eval: NLG Evaluation using GPT-4 with Better Human Alignment, arXiv 2023
>
> ---
>
> > **Q1.** *The summarization datasets differ in their abstractiveness. Did you try modifying the balancing hyperparameter $\lambda$? It seems 0.2 is good for CNNDM but probably for XSUM the value is different?*
>
> You are correct. In XSUM, where summaries are already highly abstract, it is more advantageous to prioritize the improvement of Rouge, as substantial gains in novel n-grams might not be expected. We conducted an experiment on XSUM using $\lambda$ smaller than 0.2, which places a higher weight on informativeness.
>
> |                       | ROUGE-1 | ROUGE-2 | Novel 5-gram |
> | --------------------- | ------- | ------- | ------------ |
> | BART Large            | 45.35   | 22.50    | 98.34        |
> | Discal with BART 12-3 |
> | $\lambda=0.2$            | 44.30   | 21.14   | 98.43        |
> | $\lambda=0.1$            | 44.39   | 21.19   | 98.32        |
> | $\lambda=0.0$           | 44.53   | 21.41   | 98.17        |
>
> The results indicate that decreasing lambda value leads to higher Rouge scores. Considering this trend, prioritizing the Rouge improvement is more beneficial, particularly given the high novel 5-gram score (98% of 5-grams are novel regardless of the lambda values).
>
> ---
>
> > **Q2.** *Highlighting the highest scores in the valuation tables.*
>
> Sure. We will do this.

---

### Meta-Review · Area_Chair_JgQv · 2023-09-19

**Recommendation:** 4

**Metareview:**

The paper introduces an approach, named DisCal, to improve generated summaries' abstractiveness (as determined by n-gram overlap) without compromising their informativeness (as determined by ROUGE).
DisCal offers various pseudo summaries to the student model under two supervisions: 1. the optimal pseudo summary is chosen for sequence-level distillation based on its abstractiveness and informativeness; 2. their rankings are used to guarantee that summaries with higher rankings receive higher prediction scores from the student model.

The reviewers noticed the following pros:
1. DisCal superiority over other distilled models.
2. Extensive experiments and ablation study.
3. The paper is well-structured and clearly written.

According to the changed scores, I understand that the reviewers were highly satisfied with the authors' answers in the rebuttal.

---

### Decision · Program_Chairs · 2023-10-07

**Decision:**

Accept-Findings

**Comment:**

The paper introduces an approach, named DisCal, to improve generated summaries' abstractiveness (as determined by n-gram overlap) without compromising their informativeness (as determined by ROUGE).
DisCal offers various pseudo summaries to the student model under two supervisions: 1. the optimal pseudo summary is chosen for sequence-level distillation based on its abstractiveness and informativeness; 2. their rankings are used to guarantee that summaries with higher rankings receive higher prediction scores from the student model.

The reviewers noticed the following pros:
1. DisCal superiority over other distilled models.
2. Extensive experiments and ablation study.
3. The paper is well-structured and clearly written.

According to the changed scores, I understand that the reviewers were highly satisfied with the authors' answers in the rebuttal.